# Technical Note: Monitoring of unsteady open channel flows using continuous slope-area method*

Kyutae Lee[1], Ali R. Firoozfar[2], and Marian Muste[2]

[1]Environmental Sciences Division, Oak Ridge National Laboratory, Tennessee, USA
[2]IIHR—Hydroscience & Engineering, Univ. of Iowa, Iowa, USA

*Correspondence to:* Marian Muste (marian-muste@uiowa.edu)

**Abstract.** The advent of low-cost pressure transducers capable of directly measuring water surface elevation enables continuous measurements of dynamic water surface slopes. This opens up a new possibility of dynamically monitoring unsteady flows (i.e., hysteresis) during the course of flood wave propagation. Hysteresis in this context refers to a looped stage-discharge rating caused by unsteadiness of flows. Hysteresis is monitored in this study using a continuous slope area (CSA) method, which uses Manning's equation to calculate unsteady discharges based on continuously measured water surface slopes. In the rising stage, water surface slopes become steeper than a steady water surface slope, resulting in higher discharges than steady-based discharges, while the trends are reversed in the falling stage. The CSA method applied on Clear Creek near Oxford (Iowa, USA) estimates the maximum differences of peak discharges by 30-40% while it shows sound agreements for low to medium range of discharges against USGS steady-based records. The primary cause of these differences is due to the use of a single channel bed slope in deriving Manning's roughness coefficients. The use of a single channel bed slope (conceptually equal to the water surface slopes at every stage in uniform flow conditions) causes substantial errors in estimating the channel roughness, specifically at high stages, because non-uniformities of natural channels result in varying (non-uniform) steady water surface slopes at each stage. While the CSA method is promising for dynamically tracking unsteady water surface slopes and flows in natural streams, more studies are yet needed to increase the accuracy of the CSA method in future research.

## 1 Introduction

A conventional slope area method has been used as means to estimate peak discharges based on high-water marks after large flood events, as those events are commonly rare and pose a measurement challenge in obtaining reliable sample data. The obtained data is then used to extend the upper limit of the stage-discharge rating curve (hQRC) which provides an important basis for timely flood management decisions and dissemination of imposed flood dangers to the public. The method estimates a (single value) peak discharge by surveying water surface elevation drops between upstream and downstream

---

\* This submission was written by the author(s) acting in (his/her/their) own independent capacity and not on behalf of UT-Battelle, LLC, or its affiliates or successors.

flood marks (e.g., typically high water-marks on bridge piers). The drop in water surface elevations for a uniform channel reach represents energy losses caused by bed roughness, and a peak discharge indirectly can be estimated using Manning's equation, measured cross-sections, estimated channel roughness coefficients, and friction slopes (i.e., channel bed slopes for a uniform reach) derived from field evidence.

A principle of the method has recently received renewed interest with the advent of low-cost pressure transducers capable of directly measuring water surface elevations, which enable continuous measurements of dynamic water surface slopes (Smith, et al., 2010). A continuous slope area (CSA) method is the approach that utilizes the same physical principle of the conventional slope area method; however, it uses on at minimum a pair of upstream and downstream water level sensors to continuously measure drops in water surface elevations instead of high water-marks. This enables hydrologists to

continuously estimate stream/river discharges throughout a hydrologic event. The CSA method can conceptually be classified as two approaches: original CSA and proposed CSA methods herein. The original CSA method is the approach which typically has been applied to steep slope channels in order that the effects of unsteady flows are negligible and thus one can calculate steady discharges. However, the proposed CSA method is to examine a potential of the approach in monitoring unsteady flows on mild slope channels. Henceforth, the CSA method refers to the proposed CSA method unless

stated otherwise. The CSA method would be significantly beneficial at a location where a stream/river channel has uniform cross-sections on mild slope channels while an intensity of hydrologic event is strong enough to generate substantial effects of unsteady flows.

The original CSA method was implemented by Smith et al. (2010) on the Babocomari River (Arizona, USA) in 2002. They deployed eight pressure transducers at both sides of the bank in four subsequent cross-sections. It was shown that the original

CSA method can be used to compute a continuous discharge hydrograph and to generate a steady hQRC. Stewart et al. (2012) also implemented the original CSA method at a network of sand-bedded ephemeral stream channels in southeast Arizona. They concluded that the gaging efforts succeeded in estimating discharges by comparing their estimates obtained with a sharp-crested weir in the most upstream location. The average channel bed slopes in the studies by Smith et al. (2010) and Stewart et al. (2012) were approximately 0.009 and 0.012, respectively, and the effects of unsteady flows were negligible.

Sudheer and Jain (2003) indicated that flood waves show a marked kinematic behavior when a channel bed slope is greater than 0.001. For all other cases, a variable energy slope driven by dynamic inertia and pressure forces should be considered in an analysis. The available experimental data show that the difference in discharges between steady and unsteady conditions becomes significant for low gradient channels exposed to large flow unsteadiness. Many scientists have found that the measured discharges between rising and falling limbs of the hydrograph at the same stage can vary approximately from 15%

to 40%, and similarly, the measured stages at the same discharge can range from 10% to 25% (Di Baldassarre and Montanari, 2009; Dottori et al., 2009; Faye and Cherry, 1980; Fread, 1973, 1975; Fenton and Keller, 2001; Gunawan, 2010; Herschy, 1995).

The objective of this study is therefore to examine the feasibility of the CSA method for monitoring unsteady flows by continuously measuring the change of water surface slopes during flood wave propagation. Neither the conventional slope

area method (Dalrymple and Benson, 1967) nor the original CSA method (Smith et al., 2010; Stewart et al., 2012) is intended to monitor unsteady flows; rather both are intended to compute a discharge based on steady flow assumptions. To achieve successful implementation of the CSA method, careful selections of channel reaches and measurements are important to accurately capture dynamic variations of free-surface slopes, inherent to the effects of unsteady flows. In subsequent sections, a case study applied to the USGS site (05454220) on Clear Creek near Oxford (Iowa, USA) is presented. The results from this case study will also be compared to the numerical estimations obtained with Fread's method (Fread, 1973, 1975; Lee and Muste, 2017).

## 2 Methodology

### 2.1 Governing equation

The CSA method utilizes the Manning's equation shown in Eq. (1).

$$Q = \frac{1}{n} A R^{2/3} S^{1/2} \tag{1}$$

where $Q$ = discharge, $n$ = Manning's roughness coefficient, $A$ = cross-sectional area, $R$ = hydraulic radius, and $S$ = friction slope.

It is known that the friction slope in Eq. (1) is equivalent to the water-surface slope (or the streambed slope) if the channel is steady and uniform, and this is an important basis for utilizing the method. However, given the fact that natural channels are invariably non-uniform, an application of the method has been also considered valid if energy losses from the energy gradient are properly taken into account in the calculation of friction slope as shown in Eq. (2).

$$S = \frac{h_f}{L} = \frac{\Delta h + \Delta h_v - k(\Delta h_v)}{L} \tag{2}$$

where $h_f$ = energy loss due to boundary friction in the reach, $\Delta h$ = the difference in water surface elevation at the two sections, $\Delta h_v$ = the difference in velocity head at the two sections, $k(\Delta h_v)$ = the energy loss due to contraction or expansion of the reach ($k = 0$ for contracting reaches and $k = 0.5$ for expanding reaches), and $L$ = the length of the reach.

The effects of channel non-uniformity can be minimized by computing the geometric mean of the channel conveyance, defined as $(1/n)AR^{2/3}$ in Eq. (1) at the two cross-sections. The discharge can subsequently be calculated as shown in Eq. (3).

$$Q = \sqrt{K_1 K_2 S} \tag{3}$$

where $K_1$ and $K_2$ = the channel conveyance at the two cross-sections.

## 2.2 Basic assumptions and considerations

The CSA method is subject to the same assumptions and protocols demonstrated by Dalrymple and Benson (1967) except for the fact that the effect of unsteady flows on the estimation of dynamic water surface slopes is considered as a key factor, rather than treating it as an error. Smith and others (2010) showed that the unsteadiness of flows did not significantly affect discharge calculations for their reaches due to steep channel slopes; however, they strictly attributed the effect of unsteadiness only to the local acceleration term (i.e., the last term on the right side of the momentum equation shown in Eq. (4)). However, it is important to note that unsteady effects do not solely represent the contribution from local accelerations. Unsteady flows also can incur longitudinal water depth variations in time due to the change of pressure forces as well as the change of velocities as a result of possible channel contractions or expansions. These effects are demonstrated by the second and third terms (i.e., pressure and convective accelerations, respectively) on the right side of Eq. (4). Particularly, it is known that the pressure term is the major cause of river hysteresis (Jones, 1916; Henderson 1963, 1966; Lee and Muste, 2017). Several researchers (Thomas, 1937; Posey, 1943; Gilcrest, 1950; Henderson 1963, 1966) also demonstrated that the spatial term in the second term in Eq. (4), representing water depth variations in the longitudinal direction can also be expressed by the temporal term as shown in Eq. (5). This implies that hysteresis is also a function of changing discharges in time. The contribution of this term increases for mild-slope channels with intense hydrologic events, leading to a considerable hysteresis. The CSA method therefore should capture this unsteady flow dynamics for continuous monitoring of discharges, while minimizing the effect of convective accelerations via a selection of suitable uniform reaches.

$$S = S_0 - \frac{\partial y}{\partial x} - \frac{V}{g}\frac{\partial V}{\partial x} - \frac{1}{g}\frac{\partial V}{\partial t} \tag{4}$$

where $A$ is the channel wetted cross-sectional area, $Q$ is the discharge, $y$ is the flow depth at two reference locations, $V$ is the channel mean velocity, and $S_o$ is the channel bottom slope.

$$\frac{\partial y}{\partial x} = -\frac{1}{c}\frac{\partial h}{\partial t} \tag{5}$$

where h is the measured stage at a single gaging station and c is the wave celerity.

While a successful implementation of the CSA method relies on several factors such as a selection of suitable reach, an accurate identification of Manning's roughness coefficients, and an accurate accounting of losses caused by scour, vegetation, and expansion and contraction of channel cross-sections, there are also other important factors. These include transducer clock drift leading time synchronization errors, vertical shift of sensor causing erroneous stage recordings, sediment clogging around sensors, and flow length estimation errors when the flow meanders about the thalweg during low flows (Stewart et al., 2012).

## 2.3 Selection of reach

A selection of suitable sites is crucial to ensure that governing equations on which the CSA method is based are applicable at a site. Dalrymple and Benson (1967) and ISO 1070 (1992) specified site selection requirements that a conventional slope-area method need to satisfy (See Table1). However, finding an ideal natural channel reach that meets all these requirements is difficult, so one should use their best engineering judgements in identifying suitable sites under the consideration of potential uncertainties. In this study, the USGS site (05454220) on Clear Creek near Oxford (Iowa, USA) is chosen because requirements specified in Table 1 are generally well met except for a minimum fall requirement. While small falls on mild slope channels provide ideal conditions for generating substantial hysteresis in a hQRC, authors should admit the fact that a magnitude of measurement errors could be relatively large compared with that of falls. In addition, due to limitations of assessing measurement uncertainties, the reach length requirement specified in ISO 1070 (1992) of Table 1 is not assessed. Additional benefit of choosing this site is that a USGS stream gage is located approximately 85m from the upstream deployed sensor. Therefore, USGS stream discharge records can be utilized for the calculation of Manning's roughness coefficients based on Eq. (1) because volumetric flow rates should be maintained over the short experimental reach because lateral inflows/outflows and seepages are negligible.

## 2.4 Experimental setup

A pair of pressure transducers (i.e., In-Situ Level Troll 500[1]), positioned approximately 200m apart, was deployed in 2015 as shown in Fig. 1. The pressure transducers have known accuracy of ±0.05% at 15 °C and resolution of 0.005% or better based on the product brochure. The upstream pressure transducer (PT) was installed on the left side of the bank, while the downstream PT was installed on the right side of the bank (looking downstream) to obtain a line of sights for a geodetic survey. It is recommended installing sensors on both sides of channel banks and at minimum of three cross-sectional locations to address any peculiarities in computing discharges and to increase reliability in the case of instrument malfunctions. While the calculation of discharges could be completed with only a pair of upstream/downstream sensors at two cross sections, a minimum of three cross sections (i.e., three water level sensors) is needed to obtain more reliable results. The redundancy of multiple cross sections can also help interpret stage data, so do increase confidence on discharge calculations (Smith et al., 2010; Stewart et al., 2012). Due to limited resources for this study, it is assumed that the use of two sensors can still provide the needed indication of the hysteretic behaviour.

The installed pressure transducers are encapsulated by the steel pipe casing, designed to measure a water column up to 10ft, which approximately corresponds to the bank full elevation at the site (see the top right picture in Fig. 1). After the deployment of sensors, a geodetic survey was conducted using Topcon Total Station[2] to record the water surface elevation at

---

[1] Use of trade, product, or commercial names does not imply endorsement by the authors or authors' institutions.
[2] Use of trade, product, or commercial names does not imply endorsement by the authors or authors' institutions.

each sensor tip. This is important step because the elevation at the tip of each sensor is used for the conversion of water surface elevations into a known vertical datum. To successfully accomplish this, the USGS reference mark near the stream gage station was chosen to provide a known geodetic point that can serve as reference vertical datum of the Total Station survey. Based on this reference point, each sensor elevation is converted to North American Vertical Datum of 1988 (NAVD

88). This conversion is necessary to facilitate the data interpretation based on the same reference system. In addition, cross-sections at each PT were also surveyed using this instrument (Fig. 2). Even though there are some local differences, the cross-sections have very similar geometric characteristics. These probe surveys have been repeated to ensure there was no vertical displacements of sensor tips and changes in the cross-sectional geometry during each site visit. Should the vertical elevations of sensors move during storm events, this movement may lead to the erroneous estimation of water surface slopes.

Moreover, other site conditions, such as types of bed materials and transitional elevations where a vegetation type is changing, were also identified to help interpret the results. It was observed that the channel bed was composed primarily of clay and the channel bank was covered with thick vegetation. Additionally, the spatial distance between the pressure sensors along the centerline of the channel and the water surface slope (i.e., corresponding to the channel bed slope) at low flows was surveyed when deploying the pressure transducers. The water surface slope at low flows was measured using Topcon

Total Station by taking shots on the water surface along either side of banks. The results were consistent regardless of bank sides. The surveyed slope was 0.00039, which coincides with the survey conducted by US Army Corps of Engineers (USACE). The surveyed channel bed slope is used as inputs for the calculation of Manning's roughness coefficients in the CSA method as well as for the simulation of the Fread's method. Fread's method is designed to compute unsteady discharges if a time series of stage data are given at a single gauge station. The method is developed based on a full one-

dimensional momentum and Manning's equation, and an average Root Mean Square Error (RMSE) is reported as approximately 4% (Fread, 1975). The Fread method is introduced herein to compare the CSA results with the numerical method because of a lack of validation data via direct field discharge measurements. Lee and Muste (2017) further modified the original Fread's method and used it for this study to better account for the actual geometry of cross sections. The modified method improved the accuracy associated with the estimation of channel conveyance factors and energy slopes,

which is particularly useful for a low-aspect (i.e., width to depth) ratio channel, approximately less than 30 such as the site selected in this study.

**2.5 Selection of hydrologic events**

A total of seven small to large scale events have been recorded during the measurement campaign in 2015 at the site. Figure 3 shows a full spectrum of these events. As indicated in Fig.3, the three events are selected for the demonstration of the CSA

method. The stage hydrographs for these selected three events are presented in detail in Fig. 4. For each event, the water level respectively has increased approximately 3.05 m, 2.13 m, and 1.25 m, and these correspond to peak discharges of 58.3, 22.3, and 9.5 $m^3$/sec. During the last 21 years (i.e., from 1994 to 2015) of data recording period at this site, there have been thirteen events approximately within a similar range of these scales (i.e., 9.5 $m^3$/s – 58.3 $m^3$/s), while eight events were larger

than 58.3m$^3$/sec. Interestingly, as shown in Fig. 3, water surface elevations of the last hydrologic event right after the event 3 has drastically increased for both of deployed sensors, while the USGS water level sensor located 85 m upstream from the first PT did not show any peculiarities. Unfortunately we were not able to identify the cause at the moment, but it might have been caused by a large log jammed near PT1, so subsequently increased water surface elevations downstream.

## 2.6 Estimation of the roughness coefficient

An accurate determination of the channel roughness is one of the most important factors which lead a successful use of the CSA method. An accuracy of the roughness coefficient is influenced by many factors, including geometric characteristics of bed materials, non-homogeneities of channel surface and vegetation, variations in channel shapes (i.e., expansion and contraction), obstructions, and degrees of meandering (Cowan, 1956). In addition, flow depths, seasonal vegetation changes, effects of suspended materials and bedload, and effects of deposition and scouring are also the influential factors that cause the energy losses (Chow, 1959). Moreover, it is demonstrated that effects of unsteady flows, excessive turbulences, and interactions between flood plains and main channels could also contribute to energy losses, specifically during large flood events (Trieste and Jarrett, 1987). Due to these natural complexities, it is often difficult to isolate their individual contributions to the total channel roughness (Coon, 1988).  Among these factors, the effects of vegetation condition changes (e.g., due to uprooted, inclined, or washed downstream) and/or cross-sectional changes (e.g., due to deposition, scouring, and debris jams) on the channel roughness between rising and falling phases of unsteady flows are assumed negligible in this study because of a lack of enough scientific evidence and experimental difficulties of differentiating the causes.  However, authors would like to note that those may artificially cause hysteretic effects on measured water surface slopes.

There have been extensive efforts in a hydrometric community to develop methodologies that can provide an accurate estimation of the channel roughness coefficients. Conventional practices of estimating channel roughness coefficients shown in Eq. (1) are via a) a direct estimation from known discharges and hydraulic properties; b) an indirect estimation from experimental equations (e.g., Limerinos, 1970; Bray, 1979; Jarrett, 1984; Sauer (U.S. Geological Survey, written communication), 1990); c) an indirect estimation from published n-value tables (e.g., Dalrymple and Benson, 1967; Chow, 1959; Henderson, 1966; Jarrett, 1985) or photographs of similar channels (e.g., Barnes, 1967; Aldridge and Garrett, 1973). Approach c) is generally the outcomes from either approach a) or b), and an accuracy of the method largely depends on a hydrologist's experience. Approach a) is considered the most accurate among others as measured (steady or unsteady) discharges (i.e., calibration data) can directly be used to establish a stage-*n* rating. Based on the availability of data, it may even be possible to generate a stage-*n* rating under various seasonal conditions (i.e., vegetation growing season vs. non-growing season). Once a stage-*n* rating is established, it can continuously provide accurate channel roughness information for successful implementations of the CSA method. In this study, approach a) is used based on the Manning's equation, Eq. (1) by utilizing known discharges and measured hydraulic properties such as area, hydraulic radius, and steady uniform water surface slope measured at a low flow condition (i.e., channel bed slope). Known discharges in this study are obtained from the USGS stream flow records, established through direct measurements of discharges when a flow is either steady or

unsteady. Typically, the measured unsteady discharges are corrected at USGS stream gaging stations for its unsteady effects when constructing a steady-based rating curve based on the procedures described in Rantz and others (1982). An accuracy of measured USGS discharges is known to be within 5-10% (Hirsch and Costa, 2004).

Furthermore, approach b) is also examined to compare the performance of experimental formulas with the outcomes from approach a). While there are many known experimental equations which utilizes a variety of parameters such as particle diameter, hydraulic radius, water surface slope, and /or other geometric characteristics, authors chose three simple equations that only utilize hydraulic radius and/or water surface slope. The chosen equations are shown below in Eqs. (4) to (6).

$$n = 0.104S^{0.177} \text{(Bray, 1979)} \tag{4}$$

$$n = 0.39S^{0.38}R^{-0.16} \text{(Jarrett, 1984)} \tag{5}$$

$$n = 0.11S^{0.18}R^{0.08} \text{( Sauer (U.S. Geological Survey, written communication), 1990)} \tag{6}$$

where $S$ is friction (or water surface) slope in steady uniform flow conditions.

When these experimental equations were developed, they implicitly take into account the effects of many flow retarding factors such as bank roughness, flow unsteadiness, cross-sectional irregularities, and variations in channel size. However, all these relevant effects are lumped together in deriving the equations simply because of difficulty in isolating causes.

## 2.7 Implementation Procedures

Figure 5 summarizes step-by-step implementation procedures based on the information provided hitherto. A selection of proper sites is the beginning step. Once it is determined, a series of water level sensors are installed, and a geodetic survey is conducted to ensure that measured water depths can be converted to a global vertical control datum such as NAVD 88 system. This will allow hydrologist accurately to compute the fall between water level sensors. Once water level sensors are ready for continuous recording of dynamic water surface slopes, hydrologists need to conduct a channel bed slope ($S_o$) and cross-section surveys for discharge calculations. Area ($A$) and hydraulic radius ($R$) between water level sensors can be estimated based on the surveyed cross-sections. If flow data based on field discharge measurements (e.g., USGS field measured discharges) or similar sets of data are available at a selected site, they can be used indirectly to compute the channel roughness based on Manning's equation (i.e., approach a) in Section 2.6). Surveyed channel bed slope, area, and hydraulic radius are used as inputs for these calculations. It is important to note that existing stage-discharge rating curves nearby CSA sites can alternatively be used in lieu of field measured discharges to compute the channel roughness because stage-discharge rating curves are fundamentally built upon field discharge measurements. The use of stage-discharge rating curves will allow an estimation of high resolution channel roughness coefficients for a wide range of stage variations since the curve should continuously have been calibrated for at site using regular field measurements, so authors herein recommend this approach. The Outcome from this approach is a stage-$n$ rating which specifies relationships between stages and channel roughness coefficients. If either field or rating curve based discharges are not available, other alternatives should

be used (i.e., approaches b) or c) in Section 2.6). Once done, continuous unsteady discharges can be computed at each time step by replacing the constant bed slope with the dynamic water surface slopes and utilizing the same measured area, hydraulic radius, and estimated channel roughness coefficients. Furthermore, the effects of channel non-uniformity can be minimized by computing the geometric mean of the channel conveyance based on Eq. (3).

## 3 Results and discussion

Figure 6 shows the computed variation of water surface slopes between the pressure transducers as a function of stage at the downstream PT for the three selected 2015 hydrologic events. The axis scales are set consistent across the figures to help readers compare their relative event scales. The water surface slopes are computed using the measured water surface elevations at each pressure transducer and the distance measured along the centerline of the channel between them. The results clearly confirm the existence of hysteresis in water surface slopes in the rising and falling limbs. It is found that hysteresis loop in this study rotates counter-clockwise direction as generally known as the direction of hysteresis. Hysteresis loop is indicated as the red line for rising limbs and the blue line for falling limbs in Fig. 6. Also indicated in the figure is the steady water surface slope which is equivalent to the surveyed channel bed slope (0.00039). It can be observed from this figure that as the event scale increases water surface slopes on both rising and falling limbs are steeper than the measured channel bed slope at high stages, while water surface slopes are supposed to be constant regardless of variations of stages for an ideal steady uniform reach. This implicitly indicates that even if the effects of unsteady flows are removed, water surface slopes (i.e., quasi-steady state) at high stages will still be steeper than the channel bed slope, because the channel bed slope in a uniform reach should be between rising and falling water surface slopes. This is caused by non-uniformities in channel cross-sections and non-homogeneous energy losses caused by irregular bank vegetation and bed materials which are typically unavoidable in natural streams. It is also observed that the variation of water surface slopes becomes larger as the scale of events becomes greater as shown in Fig. 6a), ranging approximately from 0.0003 to 0.0007. As the event scale increases, dynamic forces would also increase.

Figure 7 demonstrates a comparison of the channel roughness coefficients obtained with various approaches: black line represents the one based on the CSA implementation procedures (i.e., approach a)); red, blue, and violet lines each represent Bray (1979), Jarrett (1984), and Sauer (U.S. Geological Survey, written communication, 1990) methods (i.e., approach b)). It is found that there are significant differences in Manning's roughness coefficients between approach a) and selected approach b). These differences are mainly caused by the fact that site conditions that the cited equations were derived are inevitably different while some aspects such as the channel width and the channel bed slope are within the specified ranges. In practical perspective, it is therefore difficult to find appropriate experimental equations with a good accuracy. It can also be observed in this figure that the channel roughness coefficients are increasing up to the stage around 215.7 ft., and then decreasing toward higher stages. However, it is not clear yet whether this is physically a result of reduced vegetation at high

stages or erroneous roughness coefficients due to the use of a single valued channel bed slope, which are caused by channel non-uniformities and irregularities at high stages.

Figures 8 and 9 provide computed dynamic hQRCs and discharge hydrographs based on the procedures described in Section 2.7. Figure 8 does not provide with the USGS hQRC as the stage values do not coincide with the ones at the USGS gage location upstream. It is observed that both the CSA method and the modified Fread's method show substantial hysteresis while the magnitudes and patterns are different. Based on an analysis of Fig.9, it is concluded that the modified method may deviate up to 20%, while in general it is within 5-10% of ranges relative to the USGS stream discharges. These differences are primarily caused by the effects of unsteady flows. However, Fig. 9 also shows that the CSA method is only in good agreements with the USGS stream discharges at low to moderate flows. The differences can reach up to 30-40% at peak discharges, specifically for the event 1.

A sensitivity analysis is conducted to see the effects of the channel bed slope on the estimation of unsteady discharges using the event 1 as shown in Fig. 10. The computed discharges based on the measured channel slope (0.00039) are shown in Fig. 10a), while Figs. 10b) and 10c) represent the computed discharges using the assumed channel bed slope of 0.0005 and 0.0007, respectively. As the channel bed slope increases, the uncertainty near the peak discharge is decreased while it is increased for small to medium stages. This implies that the use of a single channel bed slope described in Section 2.7 in estimating the channel roughness coefficients might not be accurate enough for general applications. Due to non-uniformity and non-homogeneity of typical stream/river channels, authors herein propose the use of multiple quasi-steady water surface slopes at various segments of stages for better performance of the CSA method.

**4 Conclusions and future work**

This study provides an evidence of the CSA method's capabilities for dynamically tracking unsteady water surface slopes in natural streams, while it still needs more studies for a successful use of the method. While the magnitude of hysteresis depends on event scales and site conditions including the effects attributed to unsteady forces, the CSA method applied on Clear Creek near Oxford (Iowa, USA) estimates the maximum differences near peak discharges by 30-40% compared to the USGS steady-based records, while it shows sound agreements for low to medium range of discharges. This large difference is primarily caused by non-uniformity and non-homogeneity of natural streams/rivers. This implies that a conventional slope area method which assumes the water surface slope at the peak discharge is equivalent to the channel bed slope may not be a practical use in many cases. Furthermore, the water surface slopes at peak discharges on various magnitudes of hydrologic events are expected to vary. Therefore in order to increase the accuracy of the conventional and the proposed CSA method, a stage-segmented approach similar to a practice to build stage-discharge rating curves would be necessary for an investigation of quasi-steady state water surface slopes at various stages.

*Acknowledgements.* The authors gratefully thank Mr. Scott DeNeale of the Energy-Water Resource Systems Group in the Environmental Sciences Division at Oak Ridge National Laboratory (USA) for reviewing the manuscript.

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

**Table 1. Requirements for the selection of slope area channel reaches**

| Reference | Site selection requirements | Clear Creek case |
|---|---|---|
| Dalrymple and Benson (1967) | • Reach length ≥ 75 times of mean depth in the channel | O |
| | • Reach fall ≥ Velocity head | O |
| | • Minimum fall ≥ 0.5 ft. | X |
| ISO 1070 (1992) | • No progressive tendency of scour and fill | O |
| | • No abrupt change in the bed slope | O |
| | • Uniform cross-sections | O |
| | • Free from obstacles | O |
| | • Consistent bed material | O |
| | • Reach length: fall ≥ 10 times of uncertainty in water level differences and 20 times of uncertainty in water level measurement at one gage | Not assessed |
| | • No major tributaries | O |
| | • No overbank flows if possible | O |
| | • Consistent flow regime | O |
| | • Avoid significant curvature | O |
| | • No time lag in the reach (short distance) | O |

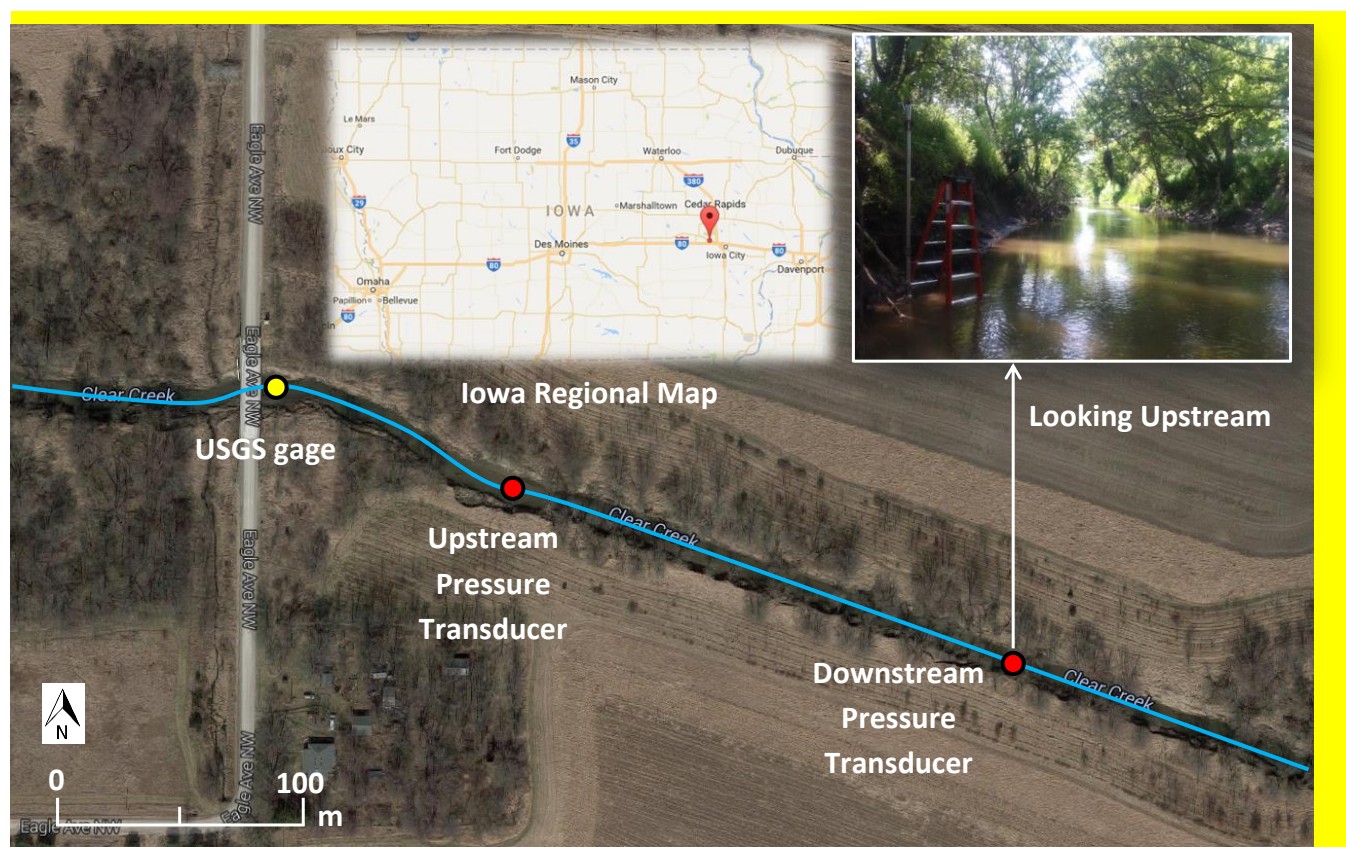

**Figure 1: Study area demonstrating a pair of pressure transducer locations at the USGS site (05454220) on Clear Creek near Oxford (Iowa, USA). Inset Iowa regional map showing the site location and background image are provided by Google Maps.**

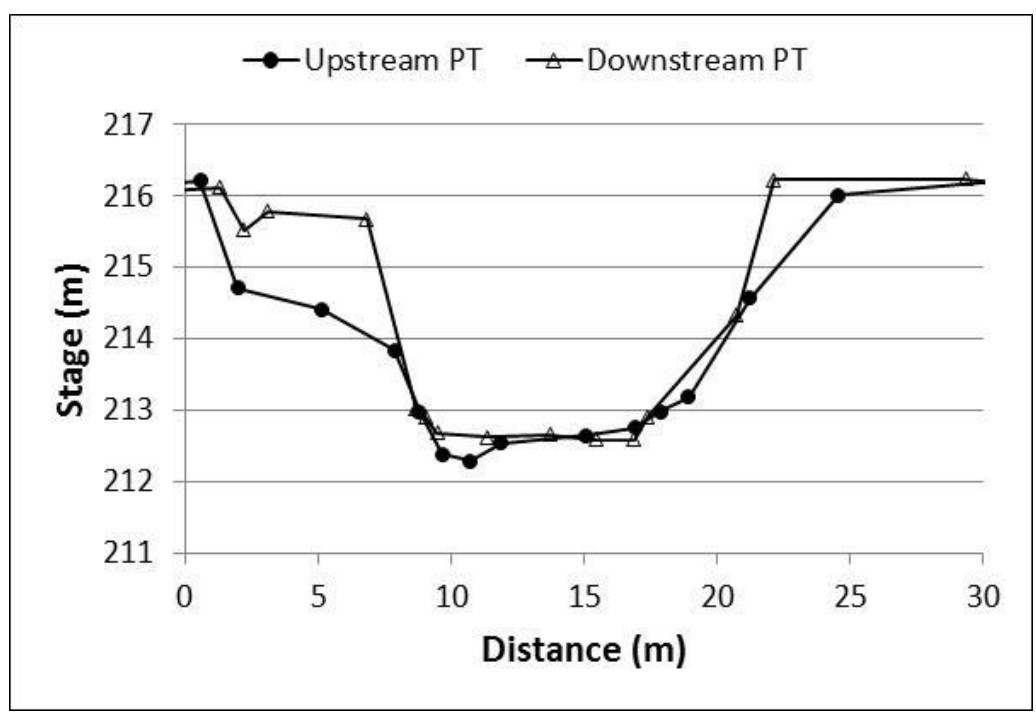

**Figure 2: Surveyed cross-sections at upstream PT and downstream PT locations. Horizontal axis represents distance from the left bank when looking upstream and vertical axis represents stages in NAVD system.**

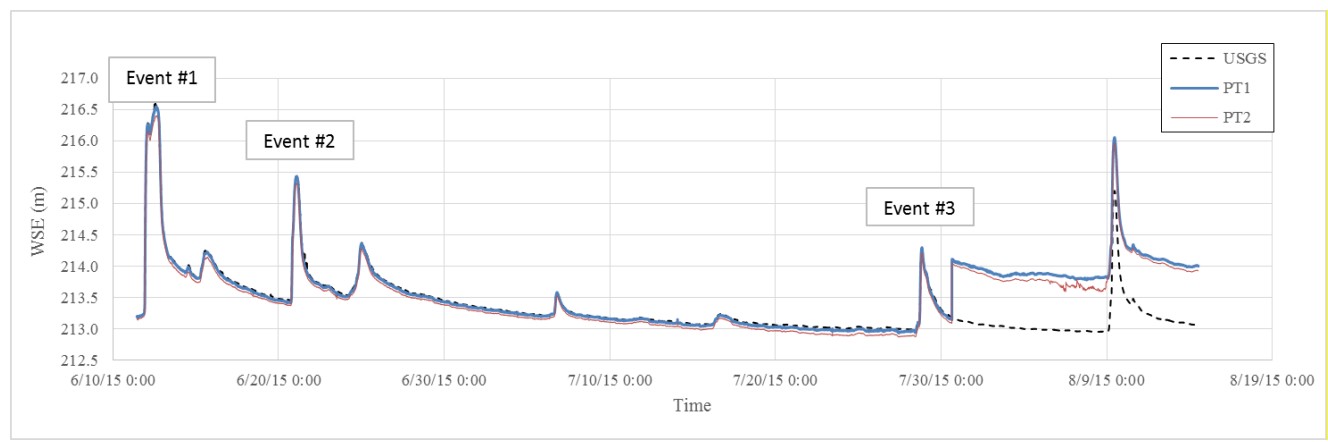

**Figure 3: A full spectrum of hydrological events during the measurement campaign. Horizontal axis shows the time stamp and vertical axis represents stages in NAVD system. The selected three events are indicated on the figure and each graph represents recorded data from USGS, PT1, and PT2 water level sensors.**

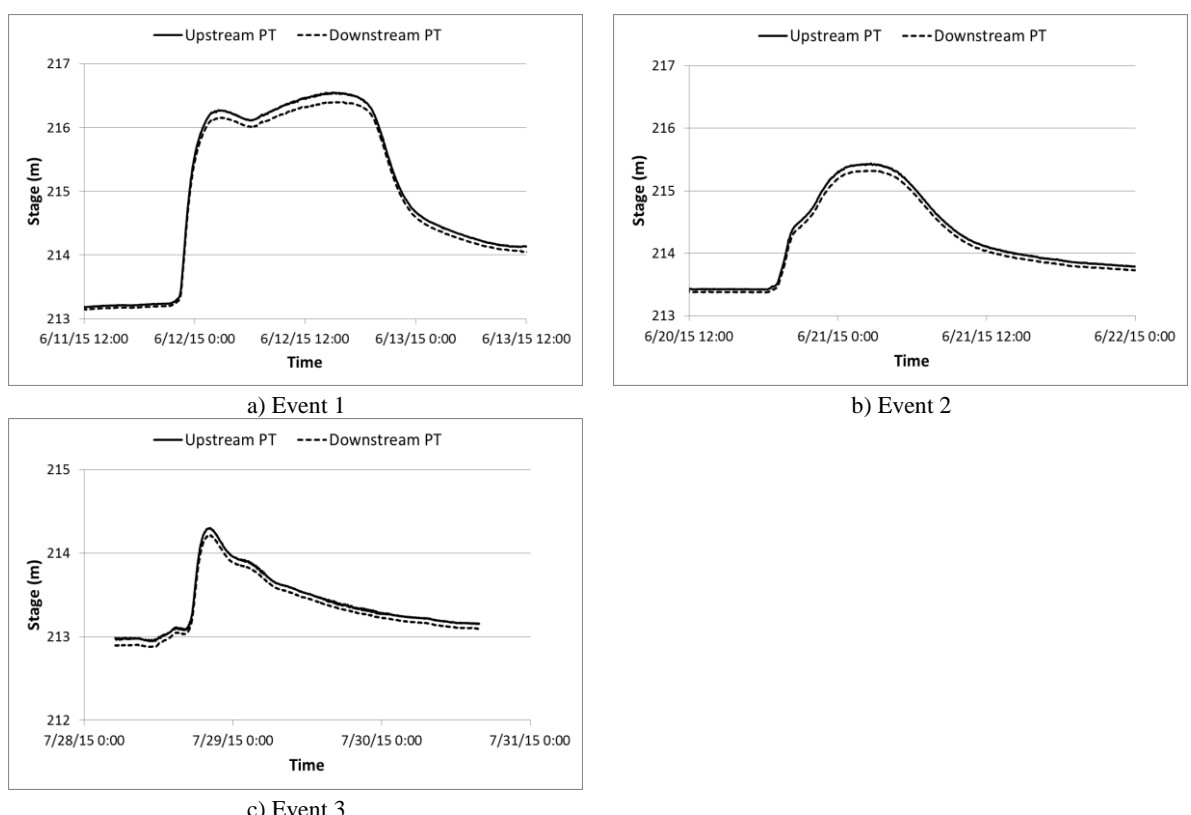

a) Event 1

b) Event 2

c) Event 3

**Figure 4: Event stage hydrographs for the selected three events. Horizontal axis represents time periods when hydrologic events were occurred and vertical axis represents stages in NAVD system.**

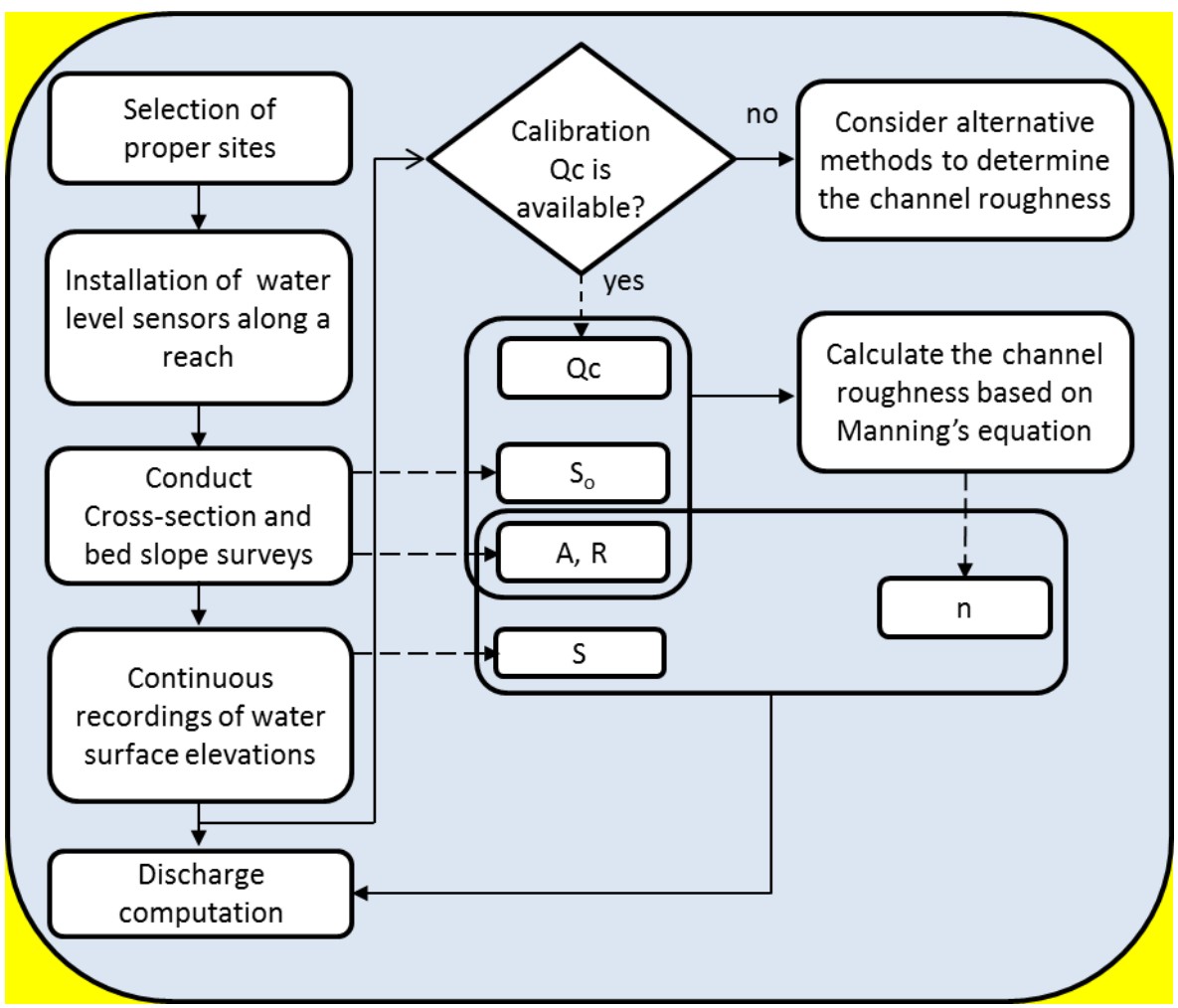

**Figure 5: A schematic diagram of the CSA method implementation procedures.**

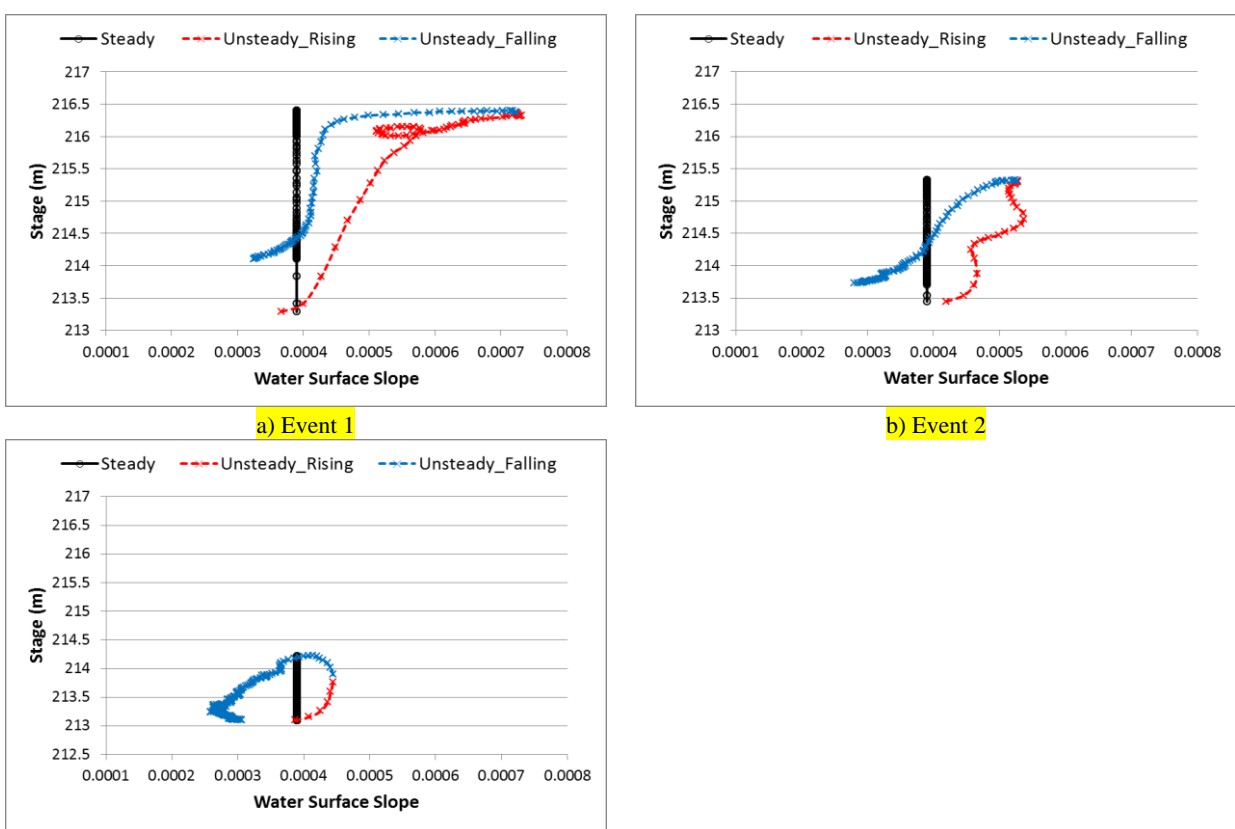

a) Event 1                                        b) Event 2

c) Event 3

**Figure 6: Measured hysteretic behaviour of water surface slopes for the selected three events. Horizontal axis represents estimated water surface slopes between transducers and vertical axis represents stages at the downstream PT.**

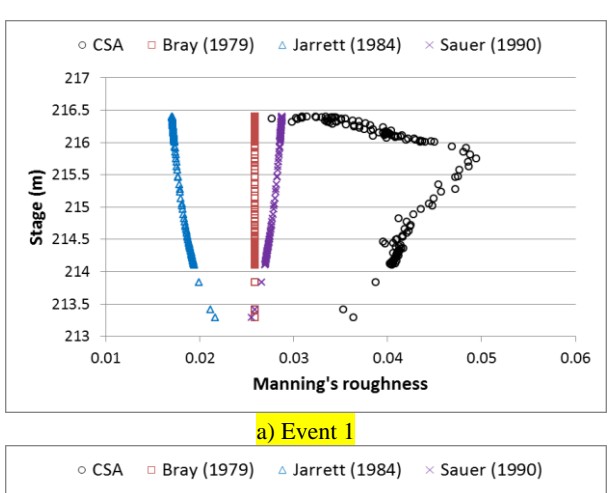

a) Event 1

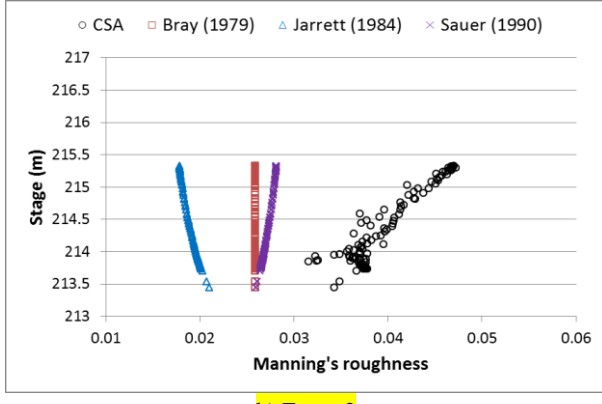

b) Event 2

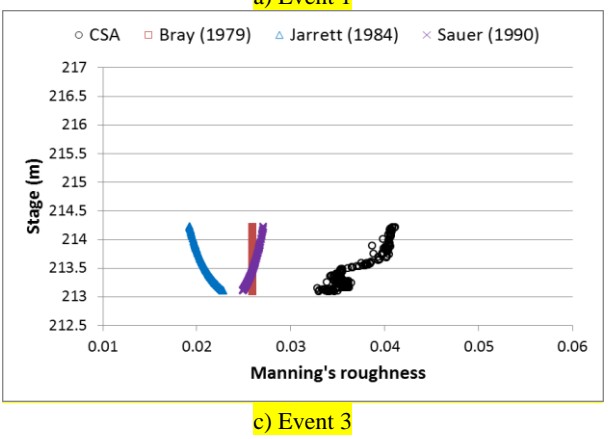

c) Event 3

**Figure 7: Comparison of the channel roughness coefficients estimated by several methods: black represents the one based on the CSA implementation procedures; red, blue, and violet data each represent Bray (1979), Jarrett (1984), and Sauer (U.S. Geological Survey, written communication, 1990) methods.**

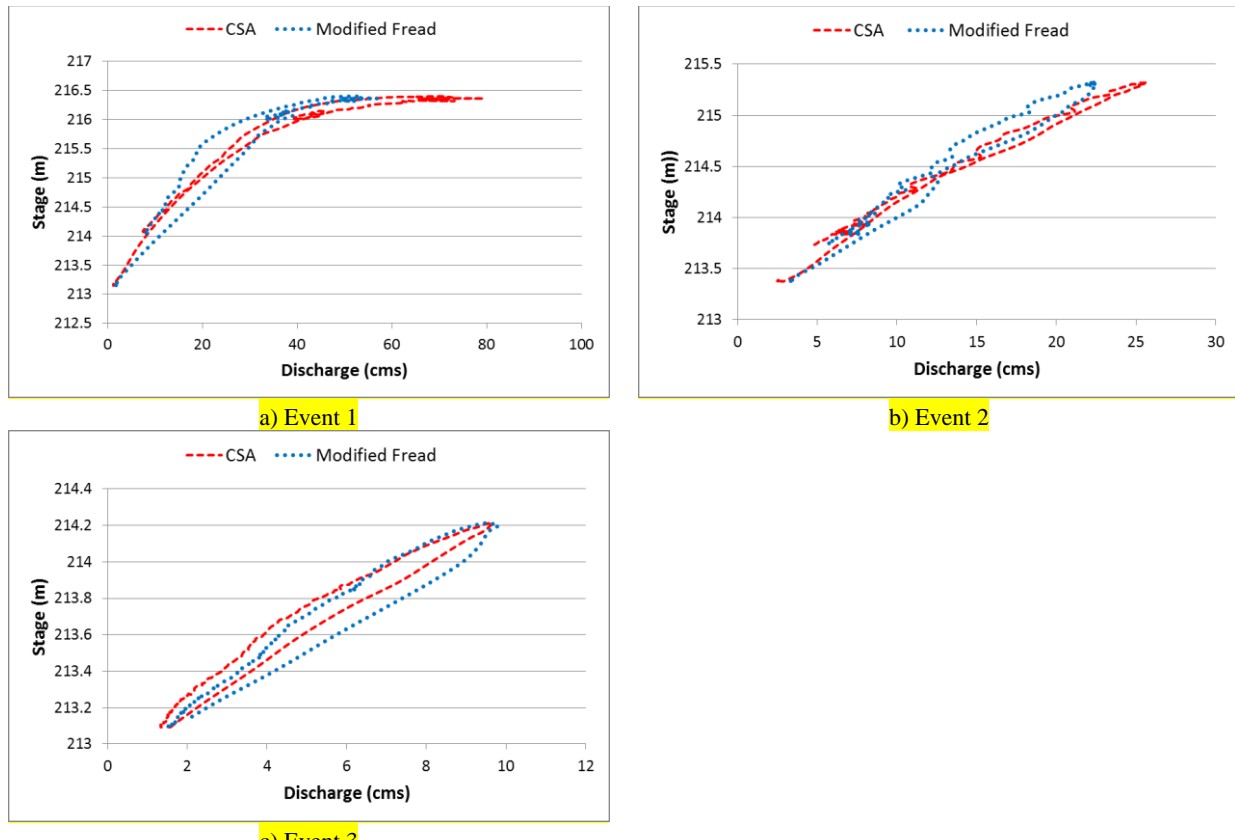

a) Event 1

b) Event 2

c) Event 3

**Figure 8: Comparison of hQRCs based on CSA and Modified Fread's methods for the selected three events. Horizontal axis represents discharges and vertical axis represents stages at the downstream PT. USGS steady hQRC is not displayed herein as stages are different as it is located 285m upstream.**

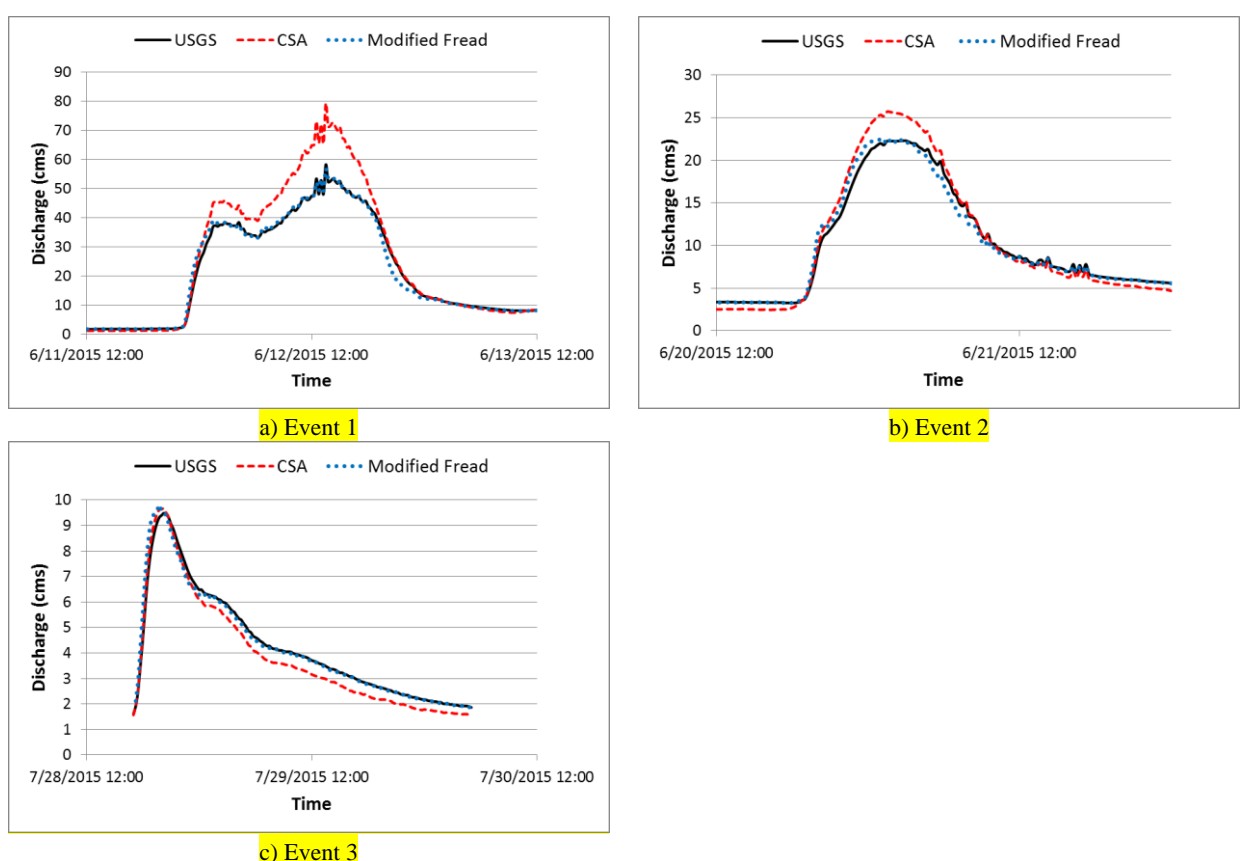

a) Event 1

b) Event 2

c) Event 3

**Figure 9: Comparison of discharge hydrographs based on the USGS, CSA, and Modified Fread's methods for the selected three events. Horizontal axis represents the time and vertical axis represents discharges.**

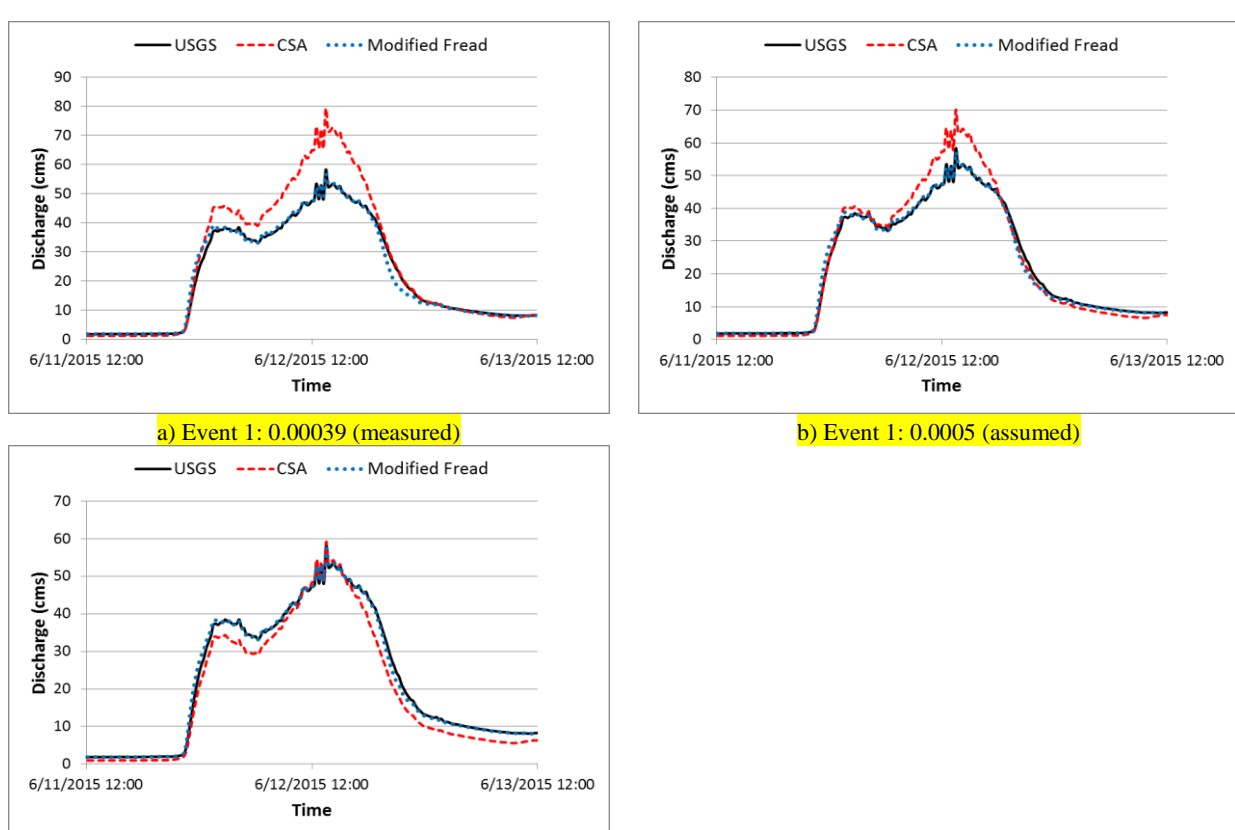

a) Event 1: 0.00039 (measured)

b) Event 1: 0.0005 (assumed)

c) Event 1: 0.0007 (assumed)

**Figure 10: Sensitivity of discharge estimates due to the change of the channel bed slopes for the event 1. The discharges are computed using the channel bed slopes of 0.00039, 0.0005, and 0.0007.**

