# Peer review of "Technical Note: Monitoring of unsteady open channel flows using continuous slope-area method\"

_Hydrology and Earth System Sciences, 2016_

## Referee Comment (RC1) · Anonymous Referee #1 · 26 Aug 2016

This review is very concise as my previous attempt to write a review was killed by the unfriendly editorial system.

The paper in itself is OK, but lacks clearness and remains too speculative. Vague words like "believe", "supposed" and "seem" indicate this.

The governing equation should be added to facilitate interpretation.

Measurement accuracy should be provided together with its consequences for the final results. The same holds for the reference of the USGS and Fread's methods. As the true discharge is not known, comparisons can only be valid if the measurement errors are taken into account.

Given the aspect ratio of the channel, not only bed roughness but also bank roughness/irregularities should be accounted for and thus addressed. Given the accessibility of the river reach characterisations of bed and bank roughness should not be a problem. Why is this information not used here? Are inferred roughness values realistic? With some information on the sediment composition, estimates regarding dynamic bed roughness can easily be made e.g. vanRijn, JHE 1984. Was there any vegetation in the domain under study? How much effect would it have? What is the rationale behind averaging the measured "unsteady slopes" knowing that the flow is subject to non-linear friction?

In its present form it merely describes the measurements done, but does not sufficiently contribute to a better understanding of the advantages of the method, and might thus not be very interesting for the readership.

It is not very convincing that the presented method is promising on the basis of the presented material.

---

## Referee Comment (RC2) · M. Perks (Referee) · 28 Aug 2016

The paper "Technical Note: Monitoring of unsteady open channel flows using continuous slope-area method" by Lee et al. seeks to adopt the use of low-cost pressure transducers to better understand the role of hysteresis in open channel flows. In its current form, the article is difficult to follow. Therefore considerable changes are required before publication can be recommended. The concept of applying the continuous slope-area method is poorly defined and described in the introduction, as is the utility of this concept. Under what conditions would applying this method be beneficial? This is the fundamental part of the manuscript so a clear explanation is required. For a Technical Note, there is a lack of detail in the Methods section. A clearly presented

[Figure]

Data Treatment section is required wherein the equations/calculations are presented. A conceptual diagram would also be beneficial to illustrate how the method is constructed and applied. A more thorough presentation of results is required, rather than simply directing the reader to the Figures.

The data used to drive the CSA method appears to be based on flow measurement, I assume collected following the development of a stage-discharge relation(?) at the USGS Clear Creek monitoring station (no information or data presented). Does this rating adequately capture both rising and falling limbs of the hydrograph? Some sensitivity analysis and discussion of this approach is required.

Specific Comments:

Page 2 Line 12 – 13: Reference required.

Page 2 Line 16: The acronym 'CSA' (first used on page 2 Line 16) is not defined in in the main body of text. This could relate to the conventional, or continuous slope area method.

Page 2 Lines 16 – 20: Strange presentation of other research. Simply stating Steward et al (2012) following their findings would suffice. No need for information about USGS/Arizona.

Page 2 Line 23: "Steep" – be specific.

Page 2 Line 24: Replace "a.k.a" with i.e.

Page 2 Line 27: "They" – who is they? If it is the series of works referenced above then their findings should be placed prior to the reference.

Page 3 Line 1: What is a "proper" reach?

Page 3 Lines 8 – 16: Useful justification for site selection. However you do not state how your chosen site meets these criteria. This information could be presented in a table.

Page 3 Lines 21 – 23: This information relating to bed slopes of sites used in other works is better suited to the introduction rather than a methods section.

Page 3 Lines 28: Assume that the Q data utilized in this research is in the form of a rating curve? This should be presented and actual method described.

Page 3 Line 29: "Cross-sectional information" is vague. Be specific.

Page 4 Lines 27 – 28: Any discussion provided by Smith et al (2010), or Stewart et al (2012) whereby the redundancy of their systems is discussed in order to back-up your use of only two sensors?

Page 4 Line 29: What pressure transducers were used? What is the associated precision and accuracy?

Page 5 Line 16: Be specific – How exactly does it compare?

Page 5 Line 19 – 20: Strangely formed sentence.

Page 5 Line 19 – 20: This is the first mention of the Fread method. How does this fit in with the experimental aims? A lack of detail is provided. If the modified Fread method is to be used then details need to be provided as the cited publication is not currently published.

Page 5 Lines 22 – 23: Small to mid-size is subjective. Catchment sizes should be given. The contributing area of Clear Creek should also be presented.

Page 5 Lines 25 – 26: Would be good to see these events placed within the context of the hydrological regime e.g. recurrence intervals.

Page 6 Lines 2 – 3: Axis information should be placed within the Figure caption.

Page 6 Lines 7 – 14: This detail, although interesting, is not related to the results. Indeed, you do not observe clockwise hysteresis so why comment on the processes driving its occurrence?

Page 6 Line 16: Use of "strong" is a subjective term – be specific.

Page 6 Line 22: Use of "very high" is a subjective term – be specific.

Page 6 Line 22: Changes in the cross-section should be presented.

Page 6 Line 25: "Sometimes not impossible" - double negative.

Page 6 Lines 26 – 27: Evidence of no major floods is provided. A Figure showing a hydrograph spanning the entire monitoring period would help place the three analysed events within the hydrological context.

Page 6 Line 30 – "Large differences" – be specific.

Page 7 Lines 30 – 32: Weak end to the conclusion. The final sentence should be more profound than being about time synchronization issues.

Figures:

General point: Appearance of all the figures and detail in the captions should be improved prior to publication.

Fig 1: A regional map as an inset would be useful to provide context. Credit to background image should be provided if appropriate.

Fig 2: Difficult to see details but at the peak stage, it looks like the steady non-uniform slope values are less that the rising and falling stage slope.

Fig 4: No useful information provided in the caption. Needs a better description.

[Figure]

---

## Author Comment (AC1) · 27 Sep 2016

**Revision Note**

The authors thank the reviewer for the suggestions and comments on how to strengthen the paper. Our specific responses to the comments are as follows:

**Anonymous Referee #1**

**1) Comments to author:**

The paper in itself is OK, but lacks clearness and remains too speculative. Vague words like "believe", "supposed" and "seem" indicate this.

**Response**

We will remove unclearness throughout the paper by providing more scientifically based approach and rationale of doing so.

**2) Comments to author:**

The governing equation should be added to facilitate interpretation.

**Response**

We will add it.

**3) Comments to author:**

Measurement accuracy should be provided together with its consequences for the final results. The same holds for the reference of the USGS and Fread's methods. As the true discharge is not known, comparisons can only be valid if the measurement errors are taken into account.

**Response**

Thank you. We will estimate the total uncertainty associated with discharge measurements when using the continuous slope area method. This estimation will be based on standard uncertainty analysis method such as ASME PTC-19.1 (2013) and GUM (1993), and will primarily account for the uncertainties associated with measured water surface slopes and Manning's roughness coefficients. In addition, an accuracy of discharges from USGS records is known to be within 5-10% (Hirsch and Costa, 2004) and an average RMS error for Fread's method is known to be approximately 4% (Fread, 1975).

*Reference*

ASME PTC19.1, "Measurement Uncertainty", American Society of Mechanical Engineers, New York, USA, 2013.

GUM., 1993. Guide to the Expression of Uncertainty in Measurement. ISBN 92-67-10188-9. BIPM, IEC, IFCC, ISO, IUPAC, IUPAP, OIML, International Organization for Standardization, Geneva, Switzerland.

Hirsch, R.M. and Costa, J.E., 2004. US stream flow measurement and data dissemination improve. Eos, 85(20), pp.197-203.

Fread, D.L., 1975. Computation of stage-discharge relationships affected by unsteady flow. JAWRA Journal of the American Water Resources Association, 11(2), pp.213-228.

**4) Comments to author:**

Given the aspect ratio of the channel, not only bed roughness but also bank roughness/irregularities should be accounted for and thus addressed. Given the accessibility of the river reach characterisations of bed and bank roughness should not be a problem. Why is this information not used here? Are inferred roughness values realistic? With some information on the sediment composition, estimates regarding dynamic bed roughness can easily be made e.g. vanRijn, JHE 1984.

**Response**

Thank you very much for this great comment. We totally agree with the concerns and issues raised by the reviewer. Conventional practices of estimating Manning's roughness coefficients are based on a) computation from experimental equations; b) selection from the published n-value table; c) comparison with photographs of channels for which n values have been computed. Since the second and third methodologies (i.e., b) and c)) are subjective and the accuracy largely depends on a hydrologist's experience, there have been a lot of efforts to estimate *n* values based on experimental equations (i.e., more objective approach, c)). As exemplified by the reviewer, we will review experimental equations that can account for bank roughness/ irregularities due to vegetated bank conditions as well as other flow retarding factors including particle diameters, cross-sectional irregularities, and variations in channel size. The review of those equations would include for example the methodologies proposed by Bray (1979), Jarrett (1984), and Sauer (1990), and the performance of those equations that can be applicable to Clear Creek conditions will be demonstrated.

However, it is also important to note that those experimental equations neglected the effect of flow unsteadiness while the *n* value is outcomes from a combination of those individual flow retarding factors, including flow unsteadiness. Coon (1998) demonstrated that the *n* value can reflect energy losses, such as those resulting from unsteady flow, extreme turbulence, and transport of suspended material and debris, that are difficult or impossible to isolate and quantify. Due to its difficulty, the effect of unsteadiness on the estimation of Manning's roughness

coefficients has typically been neglected. For example, USGS crest-stage gages are used only to measure water surface slopes at peak stages (i.e., steady discharges) for each flood event (instead of measuring whole water surface slopes during rising and falling stages). Recorded water surface slopes at different peak stages specific for each flood event can account for the effect of non-uniformity of channel conditions, but cannot account for the effects of unsteadiness. Coon (1988) even considered that measured water surface slopes obtained during rising and falling stages of a flood flow are erroneous slopes.

Given that we are now capable of measuring unsteady slopes using a pair of transducers in a continuous manner, the *n* value may be accurately estimated if field unsteady discharge data obtained during rising and falling stages (i.e., calibration data) is available. This data can be used to create stage-*n* rating under various seasonal conditions (i.e., growing season vs non-growing season). This stage-*n* rating may represent two distinct curves that reflect dynamic changes of vegetation conditions during flood wave propagation. For example, the *n* value becomes smaller on the falling stage as vegetation is already inclined toward a flow direction due to the forces exerted by a flow during the rising stage (Smith, 2010). While this is the new method of estimating the *n* value for this proposed study, the initial manuscript suggested that the value *n* be estimated using assumed steady slopes by simply averaging two unsteady slopes. This assumption comes from the fact that the value of steady water surface slopes remains between the values of two unsteady slopes at the same stage because pressure gradient term (the largest contribution factor for changing discharges) in 1D Saint-Venant equation plays a role of adding or subtracting values from the steady water surface slopes, resulting in hysteresis in stage-discharge rating curves.

As we acknowledge the reviewer's comment that this approximation lacks experimental and theoretical supports, we will change the manuscript by eliminating this approach, and introducing the suggested use of stage-*n* ratings (i.e., the use of field calibration data from USGS records along with the measured geometric data, the measured water surface slopes, and Manning's equation). While the accuracy of this approach will largely depend on the availability of calibration data, it is scientifically well based approach.

*Reference*

Bray, D.I., 1979, Estimating average velocity in gravel-bed rivers: American Society of Civil Engineers, Journal of the Hydraulics Division, v. 105, no. HY9, p. 1103-1122.

Coon, W.F., 1998. Estimation of roughness coefficients for natural stream channels with vegetated banks (Vol. 2441). US Geological Survey.

Jarrett, R.D., andPetsch, H.E., Jr., 1985, Computer program NCALC user's manual verification of Manning's roughness coefficient in channels: U.S. Geological Survey Water-Resources Investigations Report 85-4317, 27 p.

Sauer, U.S. Geological Survey, written communication, 1990 as reported in Coon, W.F., 1998. Estimation of roughness coefficients for natural stream channels with vegetated banks (Vol. 2441). US Geological Survey.

Smith, C. F., Cordova, J. T., and Wiele, S. M.: The continuous slope-area method for computing event hydrographs, US Geological Survey2328-0328, 2010.

**5) Comments to author:**

Was there any vegetation in the domain under study? How much effect would it have?

**Response**

We identified that the channel bank was covered with thick vegetation while the channel bed was composed primarily of clay. The effect of vegetation will be examined by a) the review of existing experimental equations and b) the use of field unsteady flow data from USGS records measured at different periods of time, while quantification of those effects may be limited for this study due to a small set of available data.

**6) Comments to author:**

What is the rationale behind averaging the measured "unsteady slopes" knowing that the flow is subject to non-linear friction?

Due to limited availability of measured discharge data, the simple averaging method is assumed in the original manuscript while we acknowledge the fact that the flow is subject to non-linear friction. As we demonstrated in response to comments #4, we will eliminate this approach in the revised manuscript and propose the method that can estimate the $n$ value for unsteady flow conditions by creating stage-$n$ ratings. Manning's $n$ values are often selected from tables, but can be back calculated from field discharge measurements obtained during unsteady flow conditions. Once stage-$n$ ratings are established along with measured (unsteady) water surface slopes, unsteady flow discharges can be calculated in continuous and accurate manners.

---

## Author Comment (AC2) · 27 Sep 2016

**Revision Note**

The authors thank the reviewer (Dr. Matthew Perks) for the suggestions and comments on how to strengthen the paper. Our specific responses to the comments are as follows:

**Referee #2 (M. Perks)**

**1) Comments to author:**

The paper "Technical Note: Monitoring of unsteady open channel flows using continuous slope-area method" by Lee et al. seeks to adopt the use of low-cost pressure transducers to better understand the role of hysteresis in open channel flows. In its current form, the article is difficult to follow. Therefore considerable changes are required before publication can be recommended. The concept of applying the continuous slope-area method is poorly defined and described in the introduction, as is the utility of this concept. Under what conditions would applying this method be beneficial? This is the fundamental part of the manuscript so a clear explanation is required.

**Response**

Thank you. We will modify introductions by adding better descriptions associated with the concept of applying the CSA method, utilities, and specific objectives of this research. Moreover, specific conditions such as channel bed slope (mild vs steep), vegetation conditions (heavy vs. light), and intensity of hydrologic events (i.e., rapid vs slow of water level changes) that this method would be more applicable/beneficial will be described.

**2) Comments to author:**

For a Technical Note, there is a lack of detail in the Methods section. A clearly presented Data Treatment section is required wherein the equations/calculations are presented. A conceptual diagram would also be beneficial to illustrate how the method is constructed and applied. A more thorough presentation of results is required, rather than simply directing the reader to the Figures.

**Response**

Thank you. We will add relevant equations (i.e., Manning's equation and 1D Saint Vernant equation) as well as a conceptual diagram that can help readers understand the process. More thorough presentation of results that describe how outputs (figures 6-8) are related to inputs and processes mentioned in the method section and why they are important with respect to specific objectives demonstrated in the introduction will be made.

**3) Comments to author:**

The data used to drive the CSA method appears to be based on flow measurement, I assume collected following the development of a stage-discharge relation(?) at the USGS Clear Creek monitoring station (no information or data presented). Does this rating adequately capture both

rising and falling limbs of the hydrograph? Some sensitivity analysis and discussion of this approach is required.

**Response**

Thank you. While the CSA method presented in the initial manuscript demonstrated the use of steady discharges (based on steady-based stage-discharge rating curves established by the USGS Clear Creek gage station) to estimate the Manning's roughness coefficients along with the measured cross-sections and the assumed "steady non-uniform slopes" at locations where pressure transducers are deployed. Manning's equation is used for this estimation. This USGS rating curve is steady-based stage-discharge curve, so it does not capture the effects of rising and falling limbs of the hydrograph on the estimation of discharges.

However, as we acknowledge that this previous approach lacks experimental and theoretical supports, we will change the manuscript by eliminating this approach, and introducing the suggested use of stage-$n$ ratings. Stage-$n$ ratings can be constructed by the use of field measured discharges for example from USGS historical records along with measured cross-sections, measured water surface slopes from pressure transducers (instead of assuming "steady non-uniform slopes"), and Manning's equation. While these field measured discharges are the basis of constructing steady-based stage-discharge rating curves, they implicitly captured the unsteadiness of flows. While the accuracy of this approach will largely depend on the availability of field measured discharges and their accuracy, it is scientifically well based approach. Sensitivity of computed discharges due to the uncertainties from the estimation of Manning's roughness coefficients and measured water surface slopes will be conducted and discussed.

**Specific Comments:**

**4) Comments to author:**

Page 2 Line 12 – 13: Reference required.

**Response**

The sentence will be modified as "Including more cross sections in the discharge calculation will minimize the effects of non-uniformity and will increase confidence in the computed discharges." The reference (Smith et al., 2010) will be added.

*Reference*

Smith, C. F., Cordova, J. T., and Wiele, S. M.: The continuous slope-area method for computing event hydrographs, US Geological Survey2328-0328, 2010.

**5) Comments to author:**

Page 2 Line 16: The acronym 'CSA' (first used on page 2 Line 16) is not defined in in the main body of text. This could relate to the conventional, or continuous slope area method.

**Response**

It will be corrected.

**6) Comments to author:**

Page 2 Lines 16 – 20: Strange presentation of other research. Simply stating Steward et al (2012) following their findings would suffice. No need for information about USGS/Arizona.

**Response**

It will be corrected.

**7) Comments to author:**

Page 2 Line 23: "Steep" – be specific.

**Response**

Information on Page 3 Lines 21-24 (shown below) will be moved to Page 2 Line 23 to better define steep (>0.001) channels.

"For comparison, the average channel bed slopes in the studies by Smith et al. (2010) and Stewart et al. (2012) were approximately 0.009 and 0.012, respectively, and the effects of unsteady flows were negligible in those streams. Sudheer and Jain (2003) indicated that flood waves show a marked kinematic behavior when a channel bed slope is greater than 0.001."

**8) Comments to author:**

Page 2 Line 24: Replace "a.k.a" with i.e.

**Response**

It will be replaced.

**9) Comments to author:**

Page 2 Line 27: "They" – who is they? If it is the series of works referenced above then their findings should be placed prior to the reference.

**Response**

Thank you. The sentence in Page 2 Line 27 will be re-organized prior to the reference.

**10) Comments to author:**

Page 3 Line 1: What is a "proper" reach?

**Response**

Since "proper" reach is first defined in Page 3 Lines 8-16 and the content is repeated in that section, Page 3 Line 1-2 will be removed.

**11) Comments to author:**

Page 3 Lines 8 – 16: Useful justification for site selection. However you do not state how your chosen site meets these criteria. This information could be presented in a table.

**Response**

A table will be prepared and the site selection criteria that correspond to Clear Creek conditions will be indicated.

**12) Comments to author:**

Page 3 Lines 21 – 23: This information relating to bed slopes of sites used in other works is better suited to the introduction rather than a methods section.

**Response**

Thank you for pointing this out. We agreed on the reviewer's opinion, and will move this to Page 2 Line 23 (See the response to comments #7).

**13) Comments to author:**

Page 3 Lines 28: Assume that the Q data utilized in this research is in the form of a rating curve? This should be presented and actual method described.

**Response**

Yes, Q data is obtained based on steady-based stage-discharge rating curve. The new method demonstrated in response to comments #3 will be presented while explaining how the Q data and other inputs are utilized in the proposed methodology.

**14) Comments to author:**

Page 3 Line 29: "Cross-sectional information" is vague. Be specific.

**Response**

It will be reworded by "area and hydraulic radius obtained from the surveyed cross-section".

**15) Comments to author:**

Page 4 Lines 27 – 28: Any discussion provided by Smith et al (2010), or Stewart et al (2012) whereby the redundancy of their systems is discussed in order to back-up your use of only two sensors?

**Response**

Thank you. More discussions will be provided.

**16) Comments to author:**

Page 4 Line 29: What pressure transducers were used? What is the associated precision and accuracy?

**Response**

This information will be added.

**17) Comments to author:**

Page 5 Line 16: Be specific – How exactly does it compare?

**Response**

Coincidently, the slope surveyed by USACE was also 0.00039.  So, we will be change the wording as "The surveyed slope was 0.00039, which coincides with the measurement conducted by US Army Corps of Engineers (USACE)."

**18) Comments to author:**

Page 5 Line 19 – 20: Strangely formed sentence.

**Response**

We will change the wording to be clear.

**19) Comments to author:**

Page 5 Line 19 – 20: This is the first mention of the Fread method. How does this fit in with the experimental aims? A lack of detail is provided. If the modified Fread method is to be used then details need to be provided as the cited publication is not currently published.

**Response**

The modified Fread method is to compare experimental results with the numerical method for an estimation of unsteady flows. More in-depth information will be provided while the cited paper is accepted as of Sep 19, 2016.

**20) Comments to author:**

Page 5 Lines 22 – 23: Small to mid-size is subjective. Catchment sizes should be given. The contributing area of Clear Creek should also be presented.

**Response**

Rather than defining small to mid-size, it is changed to define a low-aspect ratio channel being approximately less than 30:1 (width: depth) aspect ratio.

**21) Comments to author:**

Page 5 Lines 25 – 26: Would be good to see these events placed within the context of the hydrological regime e.g. recurrence intervals.

**Response**

We will try to find this information if available.

**22) Comments to author:**

Page 6 Lines 2 – 3: Axis information should be placed within the Figure caption.

**Response**

It will be corrected.

**23) Comments to author:**

Page 6 Lines 7 – 14: This detail, although interesting, is not related to the results. Indeed, you do not observe clockwise hysteresis so why comment on the processes driving its occurrence?

**Response**

We agree with the reviewer's comment, so this redundancy will be removed.

**24) Comments to author:**

Page 6 Line 16: Use of "strong" is a subjective term – be specific.

**Response**

We will reword the sentence as "As the event scale increases, dynamic forces would also increase.".

**25) Comments to author:**

Page 6 Line 22: Use of "very high" is a subjective term – be specific.

**Response**

It will be corrected as "the value for event 3 ranges between 0.7 and 1.3" or better wording.

**26) Comments to author:**

Page 6 Line 22: Changes in the cross-section should be presented.

**Response**

Unfortunately, the cross-section after the third event was not measured, while USGS records indicated these changes in their database.

**27) Comments to author:**

Page 6 Line 25: "Sometimes not impossible" - double negative.

**Response**

It will be corrected as "Sometimes possible".

**28) Comments to author:**

Page 6 Lines 26 – 27: Evidence of no major floods is provided. A Figure showing a hydrograph spanning the entire monitoring period would help place the three analysed events within the hydrological context.

**Response**

A figure will be added.

**29) Comments to author:**

Page 6 Line 30 – "Large differences" – be specific.

**Response**

We will replace the wording as "The computed discharge differences shown in event 1 and 2 are caused by…"

**30) Comments to author:**

Page 7 Lines 30 – 32: Weak end to the conclusion. The final sentence should be more profound than being about time synchronization issues.

**Response**

We will find the final sentence to be more profound once we have new results from the new proposed approach of estimating the Manning's roughness coefficients (see response to the comments #3).

**31) Comments to author:**

Figures:

General point: Appearance of all the figures and detail in the captions should be improved prior to publication.

Fig 1: A regional map as an inset would be useful to provide context. Credit to background image should be provided if appropriate.

Fig 2: Difficult to see details but at the peak stage, it looks like the steady non-uniform slope values are less that the rising and falling stage slope.

Fig 4: No useful information provided in the caption. Needs a better description.

**Response**

Thank you. Comments raised by the reviewer regarding figures will be taken into account in the revised manuscript.

---

## Author Response (AR1)

**Revision Note to Anonymous Referee #1**

The authors thank the reviewer for the suggestions and comments on how to strengthen the paper. Our specific responses to the comments are as follows:

**1) Comments to author:**

The paper in itself is OK, but lacks clearness and remains too speculative. Vague words like "believe", "supposed" and "seem" indicate this.

**Response**

We removed unclearness as much as possible by providing clearer approach and rationale of doing so.

**2) Comments to author:**

The governing equation should be added to facilitate interpretation.

**Response**

Added.

**3) Comments to author:**

Measurement accuracy should be provided together with its consequences for the final results. The same holds for the reference of the USGS and Fread's methods. As the true discharge is not known, comparisons can only be valid if the measurement errors are taken into account.

**Response**

Thank you. Accuracy/resolution information for this specific pressure transducer is provided. In addition, an accuracy of discharges from USGS records is known to be within 5-10% (Hirsch and Costa, 2004) and an average RMS error for Fread's method is known to be approximately 4% (Fread, 1975). This information is also included in the revised manuscript.

*Reference*

Hirsch, R. M., and Costa, J. E.: US stream flow measurement and data dissemination improve, Eos, 85(20), 197-203, 2004.

Fread, D.: Computation of stage-discharge relationships affected by unsteady flow, Journal of the American Water Resources Association, 11, 213-228, 1975.

**4) Comments to author:**

Given the aspect ratio of the channel, not only bed roughness but also bank roughness/irregularities should be accounted for and thus addressed. Given the accessibility of the river reach characterisations of bed and bank roughness should not be a problem. Why is this information not used here? Are inferred roughness values realistic? With some information on the sediment composition, estimates regarding dynamic bed roughness can easily be made e.g. vanRijn, JHE 1984.

**Response**

Thank you very much for this great comment. We totally agree with the concerns and issues raised by the reviewer. Conventional practices of estimating channel roughness coefficients are via a) a direct estimation from known discharges and hydraulic properties; b) an indirect estimation from experimental equations (e.g., Bray, 1979; Jarrett, 1984; Sauer, 1990); c) an indirect estimation from published n-value tables (e.g., Dalrymple and Benson, 1967; Chow, 1959; Henderson, 1966; Jarrett, 1985) or photographs of similar channels (e.g., Barnes, 1967; Aldridge and Garrett, 1973). Approach c) is generally the outcomes from either approach a) or b), and an accuracy of the method largely depends on a hydrologist's experience. Approach a) is considered the most accurate among others as measured (steady or unsteady) discharges (i.e., calibration data) can directly be used to establish a stage-n rating.

As exemplified by the reviewer, we also included experimental equations as alternatives of estimating channel roughness coefficients (approach b)) that can account for bank roughness/ irregularities due to vegetated bank conditions as well as other flow retarding factors including particle diameters, cross-sectional irregularities, and variations in channel size. The review of those equations includes the methodologies proposed by Bray (1979), Jarrett (1984), and Sauer (1990). The performance of those equations that can be applicable to Clear Creek conditions is demonstrated.

*Reference*

Aldridge, B. N., and Garrett, J. M.: Roughness coefficients for streams in Arizona, U.S. Geological Survey Open-File Report, 87, 1973.

Barnes, H. H.: Roughness characteristics of natural channels, U.S. Geological Survey Water-Supply Paper 1849, 213, 1967.

Bray, D. I.: Estimating average velocity in gravel-bed rivers, American Society of Civil Engineers, Journal of the Hydraulics Division, 105, 1103-1122, 1979.

Chow, V. T.: Open channel hydraulics, McGraw-Hill, New York, 1959.

Dalrymple, T., and Benson, M. A.: Measurement of peak discharge by the slope-area method, U.S. Geological Survey Techniques of Water-Resources Investigations, book 3, chap. A2, 1967.

Henderson, F. M.: Open channel flow, Macmillan, New York, 493, 1966.

Jarrett, R. D.: Hydraulics of high-gradient streams, American Society of Civil Engineers, Journal of Hydraulic Engineering, 110(11), 1519-1539, 1984.

Jarrett, R. D.: Determination of roughness coefficients for streams in Colorado, U.S. Geological Survey Water-Resources Investigations Report 85-4004, 54, 1985.

Sauer, U.S. Geological Survey, written communication, 1990 as reported in Coon, W.F., 1998. Estimation of roughness coefficients for natural stream channels with vegetated banks (Vol. 2441). US Geological Survey.

**5) Comments to author:**

Was there any vegetation in the domain under study? How much effect would it have?

**Response**

There was some vegetation as it was growing season. Since we are using existing USGS stage-discharge rating curve (i.e., discharge data) to estimate the channel roughness using Manning's equation and the curve is being calibrated regularly, the effects of seasonal vegetation should already be taken into account. It is also identified that the rating curve is shifting around with a clear pattern depending on seasons.

**6) Comments to author:**

What is the rationale behind averaging the measured "unsteady slopes" knowing that the flow is subject to non-linear friction?

We eliminated this approach in the revised manuscript and proposed more scientifically based conventional approach in estimating the channel roughness coefficients.

The authors thank the reviewer (Dr. Matthew Perks) for the suggestions and comments on how to strengthen the paper. Our specific responses to the comments are as follows:

**Referee #2 (Dr. M. Perks)**

1) **Comments to author:**

The paper "Technical Note: Monitoring of unsteady open channel flows using continuous slope-area method" by Lee et al. seeks to adopt the use of low-cost pressure transducers to better understand the role of hysteresis in open channel flows. In its current form, the article is difficult to follow. Therefore considerable changes are required before publication can be recommended. The concept of applying the continuous slope-area method is poorly defined and described in the introduction, as is the utility of this concept. Under what conditions would applying this method be beneficial? This is the fundamental part of the manuscript so a clear explanation is required.

**Response**

Thank you. We modified introductions by adding better descriptions associated with the concept of applying the CSA method, utilities, and specific objectives of this research. Moreover, specific conditions that this method would be more applicable/beneficial are also added.

2) **Comments to author:**

For a Technical Note, there is a lack of detail in the Methods section. A clearly presented Data Treatment section is required wherein the equations/calculations are presented. A conceptual diagram would also be beneficial to illustrate how the method is constructed and applied. A more thorough presentation of results is required, rather than simply directing the reader to the Figures.

**Response**

Thank you. We added the methods section with relevant equations and added implementation procedure section 2.7 with a diagram (Figure 5).

3) **Comments to author:**

The data used to drive the CSA method appears to be based on flow measurement, I assume collected following the development of a stage-discharge relation(?) at the USGS Clear Creek monitoring station (no information or data presented). Does this rating adequately capture both rising and falling limbs of the hydrograph? Some sensitivity analysis and discussion of this approach is required.

**Response**

Thank you. The reviewer is correct. Flow data are available only from USGS field measurements when they constructed a steady-state stage-discharge rating curve. The rating curve does not capture unsteadiness. Further information is provided in Section 2.7 implementation procedures.

Sensitivity of computed discharges due to the uncertainties in the measurement of channel bed slopes are also conducted and discussed as it would affect significantly the estimation of channel roughness coefficients (see Figure 10).

**Specific Comments:**

**4) Comments to author:**

Page 2 Line 12 – 13: Reference required.

**Response**

Lines 7-13 are removed.

**5) Comments to author:**

Page 2 Line 16: The acronym 'CSA' (first used on page 2 Line 16) is not defined in in the main body of text. This could relate to the conventional, or continuous slope area method.

**Response**

Corrected.

**6) Comments to author:**

Page 2 Lines 16 – 20: Strange presentation of other research. Simply stating Steward et al (2012) following their findings would suffice. No need for information about USGS/Arizona.

**Response**

Corrected.

**7) Comments to author:**

Page 2 Line 23: "Steep" – be specific.

**Response**

Information on Page 3 Lines 21-24 (shown below) is moved to better define steep (>0.001) channels.

"For comparison, the average channel bed slopes in the studies by Smith et al. (2010) and Stewart et al. (2012) were approximately 0.009 and 0.012, respectively, and the effects of unsteady flows were negligible in those streams. Sudheer and Jain (2003) indicated that flood waves show a marked kinematic behavior when a channel bed slope is greater than 0.001."

**8) Comments to author:**

Page 2 Line 24: Replace "a.k.a" with i.e.

**Response**

Corrected.

**9) Comments to author:**

Page 2 Line 27: "They" – who is they? If it is the series of works referenced above then their findings should be placed prior to the reference.

**Response**

Thank you. The sentence in Page 2 Line 27 is placed before the reference and reworded.

**10) Comments to author:**

Page 3 Line 1: What is a "proper" reach?

**Response**

The sentence is reworded as follows:

"To achieve successful implementation of the CSA method, careful selections of channel reaches and measurements are important…"

**11) Comments to author:**

Page 3 Lines 8 – 16: Useful justification for site selection. However you do not state how your chosen site meets these criteria. This information could be presented in a table.

**Response**

Table 1 is presented and the sentences are reworded.

**12) Comments to author:**

Page 3 Lines 21 – 23: This information relating to bed slopes of sites used in other works is better suited to the introduction rather than a methods section.

**Response**

Thank you for pointing this out. We agreed on the reviewer's opinion, and moved this to Introduction section (See also the response to comments #7).

**13) Comments to author:**

Page 3 Lines 28: Assume that the Q data utilized in this research is in the form of a rating curve? This should be presented and actual method described.

**Response**

Yes, Q data is obtained based on steady-based stage-discharge rating curve. A section demonstrating implementation procedures is added as Section 2.7 in the revised manuscript.

**14) Comments to author:**

Page 3 Line 29: "Cross-sectional information" is vague. Be specific.

**Response**

The sentence is removed.

**15) Comments to author:**

Page 4 Lines 27 – 28: Any discussion provided by Smith et al (2010), or Stewart et al (2012) whereby the redundancy of their systems is discussed in order to back-up your use of only two sensors?

**Response**

Thank you. More discussions are provided.

**16) Comments to author:**

Page 4 Line 29: What pressure transducers were used? What is the associated precision and accuracy?

**Response**

In-Situ Level Troll 500 is used and accuracy/resolution information is provided based on brochure.

**17) Comments to author:**

Page 5 Line 16: Be specific – How exactly does it compare?

**Response**

The slope surveyed by USACE was also 0.00039. So, we replaced "closely agrees" with "coincides".

**18) Comments to author:**

Page 5 Line 19 – 20: Strangely formed sentence.

**Response**

The sentence is reworded.

**19) Comments to author:**

Page 5 Line 19 – 20: This is the first mention of the Fread method. How does this fit in with the experimental aims? A lack of detail is provided. If the modified Fread method is to be used then details need to be provided as the cited publication is not currently published.

**Response**

The Fread method is introduced herein to compare the CSA results with the numerical method as supplemental information because direct field discharge measurements are not available. While the paper is accepted as of Sep 19, 2016, the publication process has kept being delayed. However, it will be published online in Jan-Feb, 2017 timeframe based on their promise. The citation will be updated before this article is published.

**20) Comments to author:**

Page 5 Lines 22 – 23: Small to mid-size is subjective. Catchment sizes should be given. The contributing area of Clear Creek should also be presented.

**Response**

Rather than defining small to mid-size, the wording is changed to a low-aspect ratio channel (approximately less than 30:1 (width: depth) ratio).

**21) Comments to author:**

Page 5 Lines 25 – 26: Would be good to see these events placed within the context of the hydrological regime e.g. recurrence intervals.

**Response**

Relevant information is added.

**22) Comments to author:**

Page 6 Lines 2 – 3: Axis information should be placed within the Figure caption.

**Response**

Corrected.

**23) Comments to author:**

Page 6 Lines 7 – 14: This detail, although interesting, is not related to the results. Indeed, you do not observe clockwise hysteresis so why comment on the processes driving its occurrence?

**Response**

We agree with the reviewer's comment, so this redundancy is removed.

**24) Comments to author:**

Page 6 Line 16: Use of "strong" is a subjective term – be specific.

**Response**

We reworded the sentence to "As the event scale increases, dynamic forces would also increase.".

**25) Comments to author:**

Page 6 Line 22: Use of "very high" is a subjective term – be specific.

**Response**

Removed.

**26) Comments to author:**

Page 6 Line 22: Changes in the cross-section should be presented.

**Response**

Removed.

**27) Comments to author:**

Page 6 Line 25: "Sometimes not impossible" - double negative.

**Response**

Removed.

**28) Comments to author:**

Page 6 Lines 26 – 27: Evidence of no major floods is provided. A Figure showing a hydrograph spanning the entire monitoring period would help place the three analysed events within the hydrological context.

**Response**

Figure 3 is added.

**29) Comments to author:**

Page 6 Line 30 – "Large differences" – be specific.

**Response**

Removed.

**30) Comments to author:**

Page 7 Lines 30 – 32: Weak end to the conclusion. The final sentence should be more profound than being about time synchronization issues.

**Response**

Reworded.

**31) Comments to author:**

Figures:

General point: Appearance of all the figures and detail in the captions should be improved prior to publication.

Fig 1: A regional map as an inset would be useful to provide context. Credit to background image should be provided if appropriate.

Fig 2: Difficult to see details but at the peak stage, it looks like the steady non-uniform slope values are less that the rising and falling stage slope.

Fig 4: No useful information provided in the caption. Needs a better description.

**Response**

Modified.